# Disrespect and Abuse Experienced by Women during Childbirth in Midwife-Led Obstetric Units in Tshwane District, South Africa: A Qualitative Study

**DOI:** 10.3390/ijerph17103667

**Published:** 2020-05-22

**Authors:** Refilwe Malatji, Sphiwe Madiba

**Affiliations:** Department of Public Health, Sefako Makgatho Health Sciences University, P.O. Box 215 Medunsa 0403, Pretoria 0001, South Africa; refilwemal@gmail.com

**Keywords:** South Africa, abuse, disrespective care, birthing experience, violation of rights, Maternity Care Charter

## Abstract

The disrespect and abuse (D&A) of women during childbirth is common and a great concern in midwifery-led obstetric units (MOUs) in South Africa. This paper used the seven chapters of the Respectful Maternity Care Charter as a framework to explore women’s experiences of care during childbirth and examine the occurrence of D&A during childbirth in MOUs. Five focus group interviews were conducted with postnatal women aged 18 to 45 years selected purposively from MOUs in Tshwane District in South Africa. The discussions were audio-recorded, transcribed, and analyzed using a thematic approach and NVivo11 computer software. D&A of women was common during labor and childbirth. Verbal abuse in the form of shouting, labeling, judging, and rude remarks was the common form of D&A. Some of the women were abandoned and neglected, which resulted in their giving birth without assistance. Furthermore, the midwives violated their rights and denied them care such as pain relief medication, birth companions during childbirth, and access to ambulance services. Midwives are at the center of the provision of maternity care in MOUs in South Africa. Therefore, there is a need to strengthen interventions to adopt and implement policies that promotes respectful, nonabusive care during childbirth in MOUs.

## 1. Background

The World Health Organization (WHO) identifies delivery in a health facility as an important strategy that can reduce maternal mortality, especially when the delivery is attended by skilled healthcare professionals [1,2]. Although deliveries in facilities is the most important strategy to reduce the prevalence of maternal death, some factors in low-income settings such as cultural belief, distance and transportation to health facility, costs of services, religious beliefs, and tradition of using traditional birth attendants still prevent women from using health facilities during childbirth [3,4,5]. To deliver in health facilities, research shows that women need assurance that they will be treated with dignity and respect during childbirth for them to opt to deliver in health facilities [6,7,8,9]. There is evidence that women’s previous birth experiences play a major role in their choice of current and future birth settings [6,10,11]. A negative experience during labor and childbirth damages the trust between the woman and healthcare providers and impacts on the decision regarding future delivery in a health facility [9,12,13,14]. Relevant factors include the quality of the care received, the experience of abuse and disrespect during childbirth, and the fear of maltreatment by the healthcare providers [9,15]. 

Disrespect and abuse (D&A) during childbirth are more prevalent in low-income countries, where women are often exposed to high levels of abuse and disrespectful care in public health facilities [6,8,9,16,17,18]. Experts in maternal health believe that D&A during childbirth is a barrier to the effective utilization of the skills of healthcare providers, an important indicator for Sustainable Development Goal (SDG) 5. Respectful care is a vital component to improve maternal health. 

According to the Saving Mothers’ Report for 2014–2016, women’s delay in seeking help from facilities arises from the perception that they are mistreated during childbirth [1]. Women who have experienced D&A during a previous childbirth may choose home delivery or report late for childbirth, having developed complications that could have been prevented with early presentation. There is general concern that D&A during childbirth is not only a quality care issue but is also a violation of the human rights of women [11,19,20,21]. The findings of a study conducted in Zambia linked the low quality of care to abuse during childbirth perpetrated by healthcare providers [17].

Bowser and Hill [16] describe seven categories of D&A that can take place during childbirth. These include physical abuse, nonconsented care, nondignified care, discrimination, abandonment, detention, and nonconfidential care. The D&A commonly described in studies conducted in low- and middle-income countries (LMICs) include the rudeness of staff, clinical neglect, verbal abuse, psychological abuse, and unkindness [6,8,9,12,17]. Staff attitudes reflected in behaviors such as abusive language, the denial of services, and the demonstration of an absence of compassion are some of the many barriers to the acceptance of facility-based care found in studies conducted in LMICs [10,22,23,24,25].

The chapters of the Respectful Maternity Care (RMC) Charter published by the White Ribbon Alliance are closely aligned to international mandates to create a list of seven rights that should be guaranteed to all women during pregnancy and childbirth to avert the disrespect and abuse of childbearing women [26]. The seven chapters of the RMC charter are closely aligned to the seven domains of D&A. In addition, the WHO also released a statement that emphasized the rights of women to the highest attainable standard of health, which includes the right to dignified and respectful healthcare [21]. South Africa adopted the WHO’s better birth initiative strategy of Mother and Baby-Friendly Birthing Facilities (MBFBF) in response to calls for greater attention, research, and advocacy around the maltreatment of women during childbirth. The implementation of the MBFBF strategy qualifies birthing facilities to be accredited as mother-baby friendly [21,27]. Nevertheless, D&A are common in midwifery-led obstetric units (MOUs) in primary health facilities (PHC) in South Africa [12,18,28,29].

This paper used the seven chapters of the RMC Charter as a framework to explore women’s experiences of care during childbirth and examine the occurrences of D&A during childbirth in MOUs. Improving the quality of care provided is important to increase the use of facility-based maternity care in low-income countries [30]. Research is therefore needed to understand D&A during childbirth to inform the development of appropriate and effective interventions to promote respectful care and, where appropriate, to strengthen and translate policies into meaningful action to reduce D&A [2].

## 2. Methods

### 2.1. Study Design and Setting

An exploratory qualitative research design using focus group discussions (FGDs) was conducted in MOUs in Tshwane District, Gauteng, South Africa in October 2017 using the RMC Charter as a framework (Table 1).

Tshwane District is a metropolitan municipality with a population of about 3.3 million and is one of the five districts in Gauteng Province. It has urban, peri-urban, and rural settings and has more informal settlements than other districts. The district has ten MOUs located in community health centers; seven MOUs are situated in urban areas, and three are based in peri-rural settings, servicing the rural and informal settlements. The status of women in the district mirror that of urban populations in the country. In South Africa, unemployment rates are higher amongst women 29.5% than men 25.3%, and women are likely to be employed in low-skill jobs and the informal sector [31]. Concerning the educational status of women, in functional literacy rates, women have overtaken men, but they remain less likely than men to enroll in higher degrees [32]. The MOUs provide deliveries for low-risk women at the PHC level, and they refer obstetric emergencies to the next level of care. The MOUs where the research was conducted had two referral hospitals; one is a district hospital, and the other one is a central hospital. Two of the MOUs refer directly to the central hospital due to their geographical location, the central hospital being close and there being no district hospitals in the vicinity. With regard to staffing in the MOUs, every team consists of two to five midwives, depending on the staff available for that particular MOU, and each team is led by an advanced midwife, who is a specialist midwife registered with the South African Nursing Council (SANC) [33]. Tshwane District reported the third-highest maternal mortality rate in the 2018 triennial report, which may be attributed to the large population of vulnerable groups residing in the informal settlements in the area. The district reported over 49,000 deliveries per year for the 2014–2016 triennial [34]. The MOUs and the district hospital conducted 6179 deliveries in the 2018/2019 financial year.

The participants in the study were women who had delivered in different MOUs six weeks prior to the study. One subdistrict within Tshwane District was selected for conducting FGDs with women who met the inclusion criteria. The subdistrict has four MOUs, and one was purposively selected as a setting for the FGDs, because most women who deliver in the different MOUs in the subdistrict or the hospital attend this MOU for their postnatal visits. The women were selected to participate in the FGDs with the help of the midwives in the postnatal clinic using purposive sampling. The sample was heterogeneous in nature and included women from different backgrounds, the factors taken into account being their educational level, their socioeconomic status, their age groups, and the MOUs in which they delivered [35].

### 2.2. Data Collection

The FGDs were moderated by the lead researcher (RM), hereby referred to as the moderator, and a research assistant experienced in conducting FGDs. Additionally, the moderator trained the research assistant on the objectives of the study and the focus group guide under the supervision of the second author (S.M.). The guide was developed following extensive reading on D&A during childbirth and was aligned to the rights of childbearing women. The women were asked four broad questions: (1) their satisfaction about the care provided to them during childbirth, (2) their experiences of childbirth, (3) common forms of disrespect and abuse experienced by women during childbirth, and (4) their plan to use the facilities for future deliveries. Probes and follow-up questions asked additional questions on the provision of pain medication, having a birth companion during labor, and nonconsented care. The interview guide addressed the chapters of the RMC Charter. Examples of questions are listed in Table 1.

Both the moderator and research assistant were fluent in the local languages and English and moderated the discussions using the local language (Setswana). The discussions were audio- recorded, with the permission of the participants.

Five FGDs were conducted; there were 7 to 8 women in each group, and each discussion lasted for about 60 to 90 min. The number of FGDs conducted was determined by saturation [36].

The moderator facilitated the discussions in a consulting room at the MOU, where privacy was assured. D&A are sensitive topics, and women were assured of confidentiality during the recruitment process. The moderator informed them that their participation was voluntary and that they could withdraw from the study at any stage without subsequently being penalized. All the women that were approached agreed to participate in the study and provided informed consent before the discussions were initiated. The focus groups were conducted after they had finished their medical check-ups to avoid disruption of the clinic routine. A short tool containing questions on women demographics was administered at the end of the FGDs.

### 2.3. Ethical Considerations

The Research Ethics Committee of Sefako Makgatho Health Sciences University approved the study (SMUREC/104/2017: PG). Informed consent was obtained from the women before the initiation of the data collection. The moderator-maintained confidentiality throughout the discussions by using pseudonyms.

### 2.4. Data Analysis

All the FGDs were recorded, transcribed verbatim, and translated from the local language (Setswana) into English by the lead researcher and the research assistant. Thematic analysis was done using deductive and inductive analysis to identify codes and themes from the data. [37]. The deductive approach used the seven rights or articles of the RMC Charter and the interview guide to identify priori codes. The priori codes were then used to code interviews systematically and grouped the labeled phrases into categories that were aligned with the seven domains of D&A [38]. The inductive analysis approach was used in order to generate codes directly from the data. To immerse and familiarize with the data, a few transcripts were read repeatedly by the authors and coded line by line to identify emerging initial codes. This step involved searching for meanings and identifying patterns in the data to inform the development of a codebook. The authors revised the codebook many times until they reached consensus on the definition of themes and subthemes and finalization of the codebook.

The coding of the transcripts involved the use of the qualitative data analysis software program NVivo version 11 (QSR International, Melbourne, Australia). After the authors reached consensus on the codebook, the transcripts were uploaded to NVivo 12, where further coding was done, and themes and subthemes that reflect the experiences of the women during childbirth emerged. Finally, the authors refined the emerging themes to be used in the presentation of the findings.

To ensure that the findings were a true reflection of the reality of the women during childbirth, we used a variety of methods to address credibility [35,39]. The moderator used the local language to facilitate the sessions, collected detailed field and interview notes, and used an audio recorder to collect comprehensive data for verbatim and detailed transcription. We held peer-debriefing sessions continuously throughout the research process. Both authors performed the analysis of the data to enhance reliability and reduce the effect of investigator bias. In addition, we used a computer software program for the systematic analysis of the data and a thick description of disrespectful care and abuse [35].

## 3. Results

### 3.1. Description of the Study Sample

The sample consisted of 36 women who participated in five FGDs. Their average age was 29 years, and the range was 18 to 41 years, but most were aged between 21 and 35 years. Only four women were younger than 20 years old, and seven were above 36 years. With regard to their level of education, most (27) had not completed the 12th grade, and almost all (31) were unemployed (Table 2).

### 3.2. Themes

Table 3 present the themes that emerged from the data analysis of the focus group discussions.

#### 3.2.1. Verbal Abuse and Disrespect

Every woman has the right to be treated with dignity and respect during childbirth. Respectful and compassionate care is a woman’s basic human right. The women’s narratives revealed how they were exposed to D&A during childbirth and witnessed the D&A of other women. Several subthemes emerged from this theme: women reported that nurses often shouted at them, spoke to them in harsh tones, and used abusive language when they interacted during childbirth. These subthemes are further detailed below.

##### Shouting

Verbal abuse by midwives was a common form of D&A experienced by women during childbirth. The women reported that midwives shouted at and spoke to them harshly when they failed to understand what the nurses expected from them. The midwives often failed to explain what the women must do because of language and interpretation barriers. This was particularly true for foreign and refugee women.

“The nurse shouted at me because I did not understand what they were saying, and they were laughing at me at the same time” (P 5, FGD 4).

“When the nurse spoke to me in Setswana I could not respond because I didn’t understand what she was saying and when I failed to respond she shouted and left me in pain even when the baby came” (P 4, FGD 4).

##### Rude Language

Verbal abuse of women during childbirth included the use of harsh or rude language. Women described the nurses as treating them harshly and that they used rude language when they interacted with them.

“The sister was so rude and impatient. She did not have patience with me and was hurting me” (P 5, FGD 4).

Their narratives revealed that even the cleaners were rude to them during childbirth.

“She’s a cleaner, and she does not treat women well” (P 1, FGD 3).

“The cleaner was rude to me and making me to remove my dirty sheets” (P3, FGD 5).

##### Judgmental Comments

The data revealed that the midwives were judgmental in their interactions with the women during childbirth. They told of incidents where the midwives made judgmental comments about their high parity. This was particularly an issue for foreign and refugee women.

“While busy checking my file, they (the nurses) said my child is the third and that the other two are not spaced enough. They asked why I was in such a hurry, and that I was not supposed to be pregnant. My heart was sore because they said that I abuse my children, that I should be having two children, not three” (P 1, FGD 3).

“She checked my file and said Jesus the third child. You know you are making noise! You know even our ears are painful” (P 2, FGD 1).

#### 3.2.2. Failure to Meet Professional Standards of Care

Midwives often fail to meet professional standards of care intended to address the basic needs of women during childbirth. Neglect and abandonment, delays in receiving care, failure to provide a bed, refusal to provide services, and denial to provide pain relief were the most common forms of violation of the women’s rights to the highest attainable level of healthcare and support.

##### Neglect and Abandonment

Women referred to nonresponsive midwives who often neglected them during childbirth by leaving them unattended for long periods in the labor room. They reported that they were ignored and neglected when they needed assistance during their stay in maternity care. The poor monitoring of women during labor led to some of the women delivering without assistance from the midwives.

“I was alone during the night. The nurse disappeared and I didn’t know where she was” (Pt 4, FGD 1).

“I ended up giving birth without assistance” (P 7, FGD 5).

“After the baby came, she (the nurse) was about to stitch me and I was afraid because she did not give me anything for pain, so I refused and she left me in bed with blood all over me for about two to three hours” (P 5, FGD 4).

##### Delays in Receiving Care

Ignoring women’s requests for assistance is a common type of D&A practice. Most women reported long wait times before being seen by a nurse and/or receiving care upon arrival in the facility. The delays in providing care occurred despite the severity of the pain the women were experiencing at the time.

“When I screamed for help, they just told me they are still looking for gloves” (P 4, FGD 4).

“There was another woman who came to say the baby’s head is here, she wants to give birth. They did not come to assist her. They only attended to her late when the baby’s head was out” (P 1, FGD 4).

“When I was about to deliver, they said I must not push because they were still helping another lady” (P 7, FGD 5).

##### Denied Pain Relief

Women reported that they were not offered pain medication, and often, the requests for pain relief during their labor were ignored. They were denied pain relief medication even when they cried out in pain during labor.

“They will leave you in pain and say once the baby is out the pain will go away” (P 8, FGD 3).

“When I was in pain I was crying, but they did not give me anything” (P 5, FGD 4).

The data revealed that not only was pain medication not offered, but also, women were ignored if they requested pain relief, and/or it was denied as an option for women.

“I was begging that they help me with pain medication so that they can reduce the pain, but they kept on saying there is no way that they could reduce the pain. When you deliver the baby, it will become better” (P 8, FGD 3).

##### Refusal to Provide Services

Women reported that they did not receive appropriate care acceptable to them, because the type of service was not available, or they were refused the services. The narratives revealed that, in some cases, the nurses failed to provide them with supplies even when such supplies were available. Two women reported that the nurses had refused to call ambulances for them to go to the hospital for further management.

“They said I must take my money and go to the taxis, because I cannot go by ambulance to the hospital, but there were other people whom they were transferring by ambulance” (P 1, FGD 1).

“They chased us away here. They refused to call an ambulance for me. They said I must go home and get money to go to the hospital” (P 6, FGD 5).

In South Africa, public facilities are often overcrowded, resulting in shortages of beds for women, particularly in the MOUs. As a result, some women indicated that they did not have a bed during and after childbirth.

“I was sitting on the bench the whole time when I was in labor” (P 4, FGD).

#### 3.2.3. Lack of Supportive Care

The right of women to be free from harm and ill-treatment was reported as a common form of disrespectful care by women during childbirth. They received undignified care, nonconsented care, and denial of a birth companion.

##### Undignified Care

Women reported a lack of supportive care and reluctance of midwives to help them during the delivery process. They referred to the midwives as unsympathetic, often chasing women away from the facility or refusing to assist the women during labor.

“I was bending as I was bleeding, I called the sister and when she came, blood was dropping on the floor, she said let us go and take off your dress and follow me. By then I couldn’t, my feet were not functioning well, I could not do anything, she told me not to waste her time as we must go to labor ward, she called me again” (P 4, FGD 1).

The women reported that they were discharged and told to go home during the night, because the six hours for which they were entitled to be in the health facility were over. The women felt that they received undignified care, because the midwives told them that the purpose of the discharge was to give other patients bed space, even though there were no other patients in labor.

“I was tired. They said I must go home so that I can give others space, but there were no other patients” (P 4, FGD 1).

The narratives revealed that, in some cases, the nurses failed to provide the women with supplies even when such supplies were available. Some women indicated that the midwives did not provide a bed during and after childbirth, and they sat on a bench to wait for six hours before being discharged.

“When I finished giving birth, they let me sit on the chair. Imagine the pain when they just stitched you and they tell you to sit on the chair” (P 2, FGD 1).

“As they say, they discharge you after six hours after giving birth. They do not give you a bed; they let you sit on the chair, waiting for those six hours” (P 8, FGD 4).

##### Nonconsented Care

A lack of information given to women before or during procedures and conducting vaginal examinations without their consent are common practices associated with the violation of women’s rights during childbirth. D&A also manifest through midwives performing frequent vaginal examinations without asking the women’s consent.

“Calling each other to come and check me without telling me what is happening” (P 1, FGD 4).

“But it will be nice that when you have pain they check you and tell you what is happening, other than you are being in the dark, and you become surprised when a person puts their fingers in, check you and leave you, and here you are feeling pain” (P 2, FGD 1).

The women also reported a lack of privacy in the labor wards. They referred to vaginal examinations often being conducted in nonprivate settings, having many student nurses who were there to observe the vaginal examination without the women’s consent.

“Sometimes they come with students and just check us without even telling us what they are doing or why they are doing it” (P 4, FGD 5).

##### Denied a Birth Companion

Women desired the support and presence of family members, including mothers, sisters, and spouses, during delivery who accompanied them to the facility, but they were forbidden to enter the labor ward. The deprivation of birth companions increased the feelings of being abandoned and alone during labor.

“My mother was there but they told her she should not help me that she was not meant to help me with everything. They did not chase my mother away. But she did not enter the delivery room” (P 1, FGD 2).

“He (her partner) wanted to be with me, to hold me. I could not walk. He helped me to walk, and they said, ‘Leave that woman alone and go back home’” (P 5, FGD 2).

The two women whose mothers were present during labor and delivery reported that indeed the presence of a birth companion affected their birth experiences in a positive way.

“They treated me well because my mother was there, other women who were in labor at the same time were not treated well” (P 7, FGD 1).

#### 3.2.4. Discrimination

A number of women who access health facilities in South Africa for the purposes of childbirth are the nationals of neighboring countries. These women reported that they were discriminated against by health workers due to their foreign nationality. They felt that this influenced the quality of care they received.

“For us who are from outside the country, they say ‘you Zimbabweans are burdensome, you are tiresome, and you give birth a lot’. They treat us bad” (P 5, FGD 3).

Other women felt that they were discriminated against based on high parity.

“I have four children; this is the fifth. They said ‘Why did you have the baby so quick? Magrigambas (foreigners) are irritating. You are just making babies’” (P 5, FGD 3).

“One nurse said ‘You people are annoying. You do not finish having babies. You have many children. See here, you are competing with young girls. When are you going to stop?’” (P 1, FGD 2).

#### 3.2.5. Future Utilization of Facilities

In response to questions around the future use of the facilities for childbirth, women reported that they would not use the facility for childbirth in the future. The experiences of disrespect, neglect, and abandonment by midwives during labor and childbirth affected their future utilization of the facilities.

“I will rather die than go back there” (P 2, FGD 2).

“Because of their treatment I will not come back” (P 5, FGD 4).

The experiences D&A during childbirth had a negative impact on delivery in health facilities. One woman wanted to be taught how to deliver the child by herself to avoid another facility birth.

“I want the nurses to teach me how to deliver my baby by myself at home so that I can bring the baby here after birth” (P 1, FGD 3).

A few women would return, because they had positive experiences of the birthing process.

“Because they know how to treat people right” (P 3, FGD 4).

Others would utilize the same facility for childbirth because it is close to their homes, and they would be able to have constant contact with their families for support.

“Because it is closer to my home” (P 7, FGD 1).

## 4. Discussion

This study examined the experiences of D&A during childbirth in MOUs from the perspectives of postnatal women. We used the RMC Charter as a framework to explore D&A as experienced by women in MOUs in South Africa. Maternity care in South Africa is characterized by the maldistribution of midwives, task overloads, heavy responsibilities, demoralization, a lack of motivation, poorly resourced facilities, and an ever-burgeoning demand for health services [40]. Nevertheless, some of the facilities have been accredited as mother–baby-friendly facilities in line with the WHO better birth initiative strategy [41]. The midwifery practice in South Africa is guided by a code of conduct, which stipulates that midwives must treat their patients with dignity and respect during childbirth [42]. These codes of conduct are aligned with the WHO quality of care framework for maternal and newborn care and the RMC Charter [43]

The study showed that the D&A of women in South African MOUs are common during childbirth, and women received undignified care from the midwives throughout the birthing process. These observations are consistent with reports from a recent South African study that highlighted the prevalence of D&A in all the MOUs in the same district [29]. D&A similar to that found in the current study was observed in studies conducted in other low-income settings [6,11,28,44] and in high-income settings [7].

In the current study, shouting and yelling was a common form of D&A and was often triggered by trivial things [18]. The women in yet another South African study described the mode of communication between the midwives and the women during childbirth as “forever shouting” [29]. Some research on D&A during childbirth suggests that the midwives and the women may consider the D&A to be justifiable. As such, in response to the D&A, the women most often use nonconfrontational strategies such as resigning themselves to being abused [8].

This study and others found that the attitudes and behaviors of midwives are the major contributors to D&A during childbirth [12,29]. Women were discriminated, called names, they were labeled, and they were referred to in demeaning terms. Researchers in other settings reported similar forms of adverse discrimination during childbirth [6,11,20,29]. Therefore, the rights of women to be treated equally and free from discrimination, as well as to have liberty, autonomy, self-determination, and freedom from coercion, were not observed in the present study.

The neglect and abandonment of women by midwives during childbirth was common; women recounted instances of being left alone during labor, particularly during the night, while others had to give birth without assistance. A high prevalence of abandonment and incidences of women giving birth without assistance were reported in other studies [6,7,11,12,20,45,46]. Giving birth unattended may result in complications if the baby or mother needs immediate care. The current study and others have found that disrespect, undignified care, neglect, abandonment, and abuse may be barriers to accessing healthcare in time or opting for facility-based childbirth in low-income countries [10,22,24,25].

The neglect and abandonment of women during childbirth is a violation of the RMC rights of women and regulations that govern midwife practices in South Africa. The SANC’s regulations state that, if the second stage of labor is imminent, a midwife may not leave the woman alone [47]. There is a need to refocus the practice of midwifery in South Africa to assist the midwives in adhering to the scope of their practice and the code of conduct for midwives. This will need increased efforts to address the poor staffing, task overloads, and poorly funded PHC facilities where midwives practice.

Every woman has the right to receive the highest attainable standard of healthcare and dignified and respectful care [21]. However, women in this study did not realize this right, as the denial of pain relief medication during childbirth was a common form of D&A. A recent South African study conducted in eleven MOUs in the Tshwane District found that only one facility had pain relief medication for use during labor [29]. This means that women are generally not provided with pain relief when in labor in the district. It is concerning that often midwives deny women pain relief in the belief that they must endure the labor pains and get what they deserve [48].

The violation of the human rights of women is worrisome and should not be overshadowed by the prevalent lack of resources in MOUs. For example, one woman who refused to have her episiotomy stitched without pain medication recounted her experience of being left to lie in bed with blood all over her for about three hours. Other women—particularly, foreign nationals—told of being denied ambulance services for transfer to the hospital for delivery. The failure to arrange suitable transportation for women during labor is a barrier to equitable, accessible quality maternal and child health services [49,50].

The presence of a birth companion during childbirth is one of the practices that improves maternal satisfaction; reduces the need for pain relief during labor, and leads to midwives leaning towards treating the women with dignity, compassion, and respect [26,51]. In this study and others [6,11,18,20,45,46,52], women were denied birth companions and were not informed about their right to the presence of a birth companion of their choice during childbirth. Additionally, nonconsented care that women experienced is a direct result of the lack of information during childbirth. Consistent with other studies, women did not have procedures or the labor progress explained to them and did not consent to frequent and excessive procedures [6,53]. The women’s rights to privacy were not realized for some women during childbirth whose examinations were performed in nonprivate settings.

## 5. Limitations

The qualitative nature of the study limits the ability to make broader generalizations to women who deliver in MOUs in the Tshwane District. The study did not exam the perspectives of the midwives and other health professionals on disrespectful care; therefore, their views are not reflected here and limits the ability of the findings to give a true picture of D&A in MOUs in the district and the province. One other limitation is that the study did not collect information on the characteristics of the healthcare facilities in terms of resources and the number of midwives to corroborate the statements made by the women about the lack of delivery beds. The strength of the study is the use of the RMC Charter as a framework to assess D&A as the outcome of the analysis and points to more than just the common occurrence of D&A but, also, to the violation of the rights of women during childbirth.

## 6. Conclusions

This study has highlighted that D&A in childbirth are common and that women received undignified care from midwives. Disrespectful care happened despite South Africa having adopted the better-birth initiative strategy (MBFBF). The promotion of respectful maternity care requires training on respectful care and a change in attitude, as well as the strengthening of the professional ethics training of midwives, to embed humane clinical care into routine birthing care.

## Figures and Tables

**Table 1 ijerph-17-03667-t001:** Questions from the interview guide. RMC: Respectful Maternity Care.

Rights of Women from the RMC Charter	Questions
Freedom from harm and ill-treatment.	How satisfied are you with the care you received from the health providers during labor and childbirth?
Right to information, informed consent and refusal, and respect for her choices and preferences, including the right to her choice of companionship during maternity care.	Did the midwife inform you about the right to have a birth companion present during delivery, and were you allowed to have one?
Privacy and confidentiality.	How did the midwife maintain your privacy during labor and childbirth?
To be treated with dignity and respect.	What were your birthing experiences regarding the care you received during labor and childbirth? What kind of support are women provided during labor and delivery?
Right to equality, freedom from discrimination, and equitable care.	What are the common forms of disrespect and abuse that women experience during labor and childbirth? Were you or did you witness other women being discriminated by the midwife during labor and childbirth?
Right to timely healthcare and to the highest attainable level of health.	How did the midwife help you to cope with pain during labor and childbirth?
	What informed your decision to choose to give birth in this facility, and would you choose to deliver in this facility again?

**Table 2 ijerph-17-03667-t002:** Sociodemographic characteristics of the participants.

Characteristics		Number	Percentages
Age	<20	4	13.9
	21–35	25	66.7
	>35	7	19.4
Education	Secondary education	27	75
	Completed Grade 12 and above	9	25
Employment status	Employed	5	13.9
	Unemployed	31	86.1
Parity	First pregnancy	7	19.4
	Second or third pregnancy	21	58.3
	Fourth pregnancy	5	13.8
	More than 4 pregnancies	2	5.5
Marital status	Single	25	69.4
	Married	10	27.7
	Unspecified	1	2.7
Companion	Accompanied to the facility	25	69.4
	Wanted to have a companion	21	58.3
	Had a companion	8	22.2

**Table 3 ijerph-17-03667-t003:** Themes.

Human Rights Chapter	Themes	Subthemes
To be treated with dignity and respect	Verbal abuse and disrespect	Shouting Rude language Judgmental comments
Highest attainable level of healthcare and continuous support	Failure to meet professional standards of care	Neglect and abandonment Delays in receiving care Failure to provide a bed Refusal to provide services Denied pain relief
Information, informed consent, and respect for her choices and preferences	Lack of supportive care	Undignified care Nonconsented care Denial of birth companion
Equality, freedom from discrimination, and equitable care	Discrimination	Discrimination based on nationality Discrimination based on high parity
	Future utilization of facilities

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
