# Peer review of "Disrespect and Abuse Experienced by Women during Childbirth in Midwife-Led Obstetric Units in Tshwane District, South Africa: A Qualitative Study"

_ijerph, 2020, doi:10.3390/ijerph17103667_

Round 1

Reviewer 1 Report

Many thanks for the opportunity of revising this paper. Maternal health is an important public and global health issue and this article explores respectful care for women during childbirth in obstetric units in South Africa. It also fits general audience of this journal.

Some comments though.

Introduction

The sentence “some factors still prevent and discourage women from using health facilities during childbirth” (and statements to follow) should be limited to “specific contexts”, where cultural drivers lead to this behavior. Indeed, in high-income contexts delivery outside health facilities does not happen (or it’s a rare event), irrespectively of D&A.

Methods

Information about Tshwane District can add more data on the proportion of women in the population, as well as the position of women in all aspects of society. The second point might be deepened in the discussion section, if authors prefer.

Authors specified that the district recored 101 over 49 000 deliveries per year for the 2014-2016 triennial; in my opinion, it could be useful report how many of those occurred at home or in health facilities, if available. Similarly, are volumes of activity of MOU available? I mean the number of births per year. Even if this a qualitative study, this might give you a more accurate idea of the subject of the article itself.

Discussion

The enrolled obstetric units are midwife-led, so I understand that they have no medical staff and, just an assumption, only “normal” (lower groups of Robson classification?) childbirth are performed in such settings. If compared with physician-led obstetric units, is it possible to assume any difference on the basis of literature? Is there a need of further research?

Limitation

This section is very poor and should be strengthened.

Author Response

We thank the reviewer for the comments. The response to the comments are outlined in the letter and also highlighted in the relevant sections of the manuscript.

Introduction

The sentence “some factors still prevent and discourage women from using health facilities during childbirth” (and statements to follow) should be limited to “specific contexts”, where cultural drivers lead to this behavior. Indeed, in high-income contexts delivery outside health facilities does not happen (or it’s a rare event), irrespective of D&A.

Response: We added text to highlight some factors in low income settings such as cultural belief, distance to health facilities, tradition that still prevent women from using health facilities during childbirth

Methods

Information about Tshwane District can add more data on the proportion of women in the population, as well as the position of women in all aspects of society.

Response: The status of women in Tshwane District mirror that of the urban populations in other urban settings in the country.

The second point might be deepened in the discussion section, if authors prefer

Response: We did not discuss the issues around the position of women in the society since the comments from the second reviewer was that the manuscript is too long and we felt that adding other text would make the manuscript even longer.

Authors specified that the district recorded 101 over 49 000 deliveries per year for the 2014-2016 triennial; in my opinion, it could be useful to report how many of those occurred at home or in health facilities, if available. The confidential inquiry into maternal and neonatal death.

Response: Level of care by category of maternal death, the report on the confidential inquiry into maternal deaths in South Africa shows that most maternal deaths in the country occur in district, regional, and tertiary hospitals more than in MOUs and at home. The regional hospitals serve as referrals from district hospitals and in some cases from MOUs. This explains the low levels of maternal deaths in MOUs since all high risk pregnancies are manage in secondary levels of care. Unfortunately, the data is not broken down to the level of Provinces and district.

Similarly, are volumes of activity of MOUs available? I mean the number of births per year. Even if this a qualitative study, this might give you a more accurate idea of the subject of the article itself.

Response: According to the DHIS, the MOUs collectively and the district hospital conducted 6179 deliveries in the 2018/2019 financial year in the District

Discussion

The enrolled obstetric units are midwife-led, so I understand that they have no medical staff and, just an assumption, only “normal” (lower groups of Robson classification?) childbirth are performed in such settings. If compared with physician-led obstetric units, is it possible to assume any difference on the basis of literature? Is there a need of further research?

Response: Literature shows that women who deliver in hospital experience less disrespect and abuse. This is attributed to high prevalence of physicians, advanced skills and training of midwives, and the need for constant monitoring of the high risk patient.

Limitations

This section is very poor and should be strengthened.

Response: We added a paragraph to highlight other limitations of the study.

Reviewer 2 Report

The study is a very interesting one, in line with other studies, not for that reason of less merit, it should be done more frequently in western media as a method of quality improvement in maternity facilities.

I find two drawbacks its excessive length and that opinions are not quantified, which would make the work more robust.
All this is not an obstacle to its publication.

I would suggest the authors to reduce the length of the text.

Author Response

We thank the reviewer for the comments. 

The study is a very interesting one, in line with other studies, not for that reason of less merit, it should be done more frequently in western media as a method of quality improvement in maternity facilities.

I find two drawbacks its excessive length and that opinions are not quantified, which would make the work more robust. All this is not an obstacle to its publication.

I would suggest the authors to reduce the length of the text.

Response: We reduced the length of the manuscript without affecting the merits of the study. For example, we reduced the number of quotations were it was possible.
